# Decentralized Riemannian Conjugate Gradient Method on the Stiefel Manifold

**Jun Chen**[1], **Haishan Ye**[2,3], **Mengmeng Wang**[1], **Tianxin Huang**[1]
**Guang Dai**[3], **Ivor W. Tsang**[4,5], **Yong Liu**[1]*
[1]Zhejiang University  [2]Xi'an Jiaotong University  [3]SGIT AI Lab, State Grid Corporation of China
[4]CFAR and IHPC, Agency for Science, Technology and Research  [5]SCSE, NTU
{junc,mengmengwang,21725129}@zju.edu.cn, yehaishan@xjtu.edu.cn
{guang.gdai,ivor.tsang}@gmail.com, yongliu@iipc.zju.edu.cn

## Abstract

The conjugate gradient method is a crucial first-order optimization method that generally converges faster than the steepest descent method, and its computational cost is much lower than that of second-order methods. However, while various types of conjugate gradient methods have been studied in Euclidean spaces and on Riemannian manifolds, there is little study for those in distributed scenarios. This paper proposes a decentralized Riemannian conjugate gradient descent (DRCGD) method that aims at minimizing a global function over the Stiefel manifold. The optimization problem is distributed among a network of agents, where each agent is associated with a local function, and the communication between agents occurs over an undirected connected graph. Since the Stiefel manifold is a non-convex set, a global function is represented as a finite sum of possibly non-convex (but smooth) local functions. The proposed method is free from expensive Riemannian geometric operations such as retractions, exponential maps, and vector transports, thereby reducing the computational complexity required by each agent. To the best of our knowledge, DRCGD is the first decentralized Riemannian conjugate gradient algorithm to achieve global convergence over the Stiefel manifold.

## 1 Introduction

In large-scale systems such as machine learning, control, and signal processing, data is often stored in a distributed manner across multiple nodes and it is also difficult for a single (centralized) server to meet the growing computing needs. Therefore, the decentralized optimization has gained significant attention in recent years because it can effectively address the above two potential challenges. In this paper, we consider the following distributed smooth optimization problem over the Stiefel manifold:

$$\min \frac{1}{n} \sum_{i=1}^{n} f_i(x_i),$$

$$\text{s.t. } x_1 = \cdots = x_n, \quad x_i \in \mathcal{M}, \quad \forall i = 1, 2, \ldots, n, \tag{1}$$

where $n$ is the number of agents, $f_i$ is the local function at each agent, and $\mathcal{M} := \text{St}(d, r) = \{x \in \mathbb{R}^{d \times r} | x^\top x = I_r\}$ is the Stiefel manifold ($r \leq d$) (Zhu, 2017; Sato, 2022). Many important large-scale tasks can be written as the optimization problem (1), e.g., the principle component analysis (Ye & Zhang, 2021), eigenvalue estimation (Chen et al., 2021), dictionary learning (Raja & Bajwa, 2015), and deep neural networks with orthogonal constraint (Vorontsov et al., 2017; Huang et al., 2018; Eryilmaz & Dundar, 2022).

The decentralized optimization has recently attracted increasing attention in Euclidean spaces. Among the methods explored, the decentralized (sub-)gradient method stands out as a straightforward way combining local gradient descent and consensus error reduction (Nedic & Ozdaglar, 2009; Yuan et al., 2016). Further, in order to converge to a stationary point (i.e., exact convergence) with

---

*Corresponding author

fixed step size, various algorithms have considered the local historical information, e.g., gradient tracking algorithm (Qu & Li, 2017; Yuan et al., 2018), primal-dual framework (Alghunaim et al., 2020), EXTRA (Shi et al., 2015), and ADMM (Shi et al., 2014; Aybat et al., 2017), when each local function is convex.

However, none of the above studies can solve the problem (1) since the Stiefel manifold is a non-convex set. Based on the viewpoint of Chen et al. (2021), the Stiefel manifold is an embedded sub-manifold in Euclidean space. Thus, with the help of Riemannian optimization (i.e., optimization on Riemannian manifolds) (Absil et al., 2008; Boumal et al., 2019; Sato, 2021), the problem (1) can be thought as a constrained problem in Euclidean space. The Riemannian optimization nature brings more challenges for consensus construction design. For instance, a straightforward way is to take the average $\frac{1}{n}\sum_{i=1}^{n} x_i$ in Euclidean space. However, the arithmetic average does not apply to the Riemannian manifold because the arithmetic average of points can be outside of the manifold. To address this problem, Riemannian consensus method has been developed (Shah, 2017), but it needs to use an asymptotically infinite number of consensus steps for convergence. Subsequently, Wang & Liu (2022) combined the gradient tracking algorithm with an augmented Lagrangian function to achieve the single step of consensus. Recently, Chen et al. (2021) proposed a decentralized Riemannian gradient descent algorithm over the Stiefel manifold, which also requires only the finite step of consensus to achieve the convergence rate of $\mathcal{O}(1/\sqrt{K})$. Simultaneously, the corresponding gradient tracking version was presented to reach a stationary point with the convergence rate of $\mathcal{O}(1/K)$. On this basis, Deng & Hu (2023) replaced retractions with projection operators, thus establishing a decentralized projected Riemannian gradient descent algorithm over the compact submanifold to achieve the convergence rate of $\mathcal{O}(1/\sqrt{K})$. Similarly, the corresponding gradient tracking version also achieved the convergence rate of $\mathcal{O}(1/K)$.

In this paper, we address the decentralized conjugate gradient method on Riemannian manifolds, which we refer to as the decentralized Riemannian conjugate gradient descent (DRCGD) method. In essence, the conjugate gradient method is an important first-order optimization method, which generally converges faster than the steepest descent method, and its computational cost is much lower than that of second-order methods. As well, conjugate gradient methods are highly attractive for solving large-scale optimization problems (Sato, 2022). Recently, Riemannian conjugate gradient methods have been studied, however, expensive operations such as parallel translations, vector transports, exponential maps, and retractions are required. For instance, some studies use a theoretical approach, i.e., parallel translation along the geodesics (Smith, 1995; Edelman & Smith, 1996; Edelman et al., 1998), which hinders the practical applicability. More generally, other studies utilize a vector transport (Ring & Wirth, 2012; Sato & Iwai, 2015; Zhu, 2017; Sakai & Iiduka, 2020; 2021) or inverse retraction (Zhu & Sato, 2020) to simplify the execution of each iteration of Riemannian conjugate gradient methods. Nonetheless, there is still room for computational improvements.

This paper focuses on designing an efficient Riemannian conjugate gradient algorithm to solve the problem (1) over any undirected connected graph. Our contributions are summarized as follows:

1. We propose a novel decentralized Riemannian conjugate gradient descent (DRCGD) method whose global convergence is established under an extended assumption. It is the first Riemannian conjugate gradient algorithm for distributed optimization.

2. We further develop the projection operator for search directions such that the expensive retraction and vector transport are completely replaced. Therefore, the proposed method is retraction-free and vector transport-free, and achieves the consensus of search directions, giving rise to an appealing algorithm with low computational cost.

3. Numerical experiments are implemented to demonstrate the effectiveness of the theoretical results. The experimental results are used to compare the performance of state-of-the-art ones on eigenvalue problems.

## 2 PRELIMINARIES

### 2.1 NOTATION

The undirected connected graph $G = (\mathcal{V}, \mathcal{E})$, where $\mathcal{V} = \{1, 2, \cdots, n\}$ is the set of agents and $\mathcal{E}$ is the set of edges. When $W$ is the adjacency matrix of $G$, we have $W_{ij} = W_{ji}$ and $W_{ij} > 0$ if an

edge $(i, j) \in \mathcal{E}$ and otherwise $W_{ij} = 0$. We use $\mathbf{x}$ to denote the collection of all local variables $x_i$ by stacking them, i.e., $\mathbf{x}^\top := (x_1^\top, x_2^\top, \cdots, x_n^\top)$. Then define $f(\mathbf{x}) := \frac{1}{n} \sum_{i=1}^n f_i(x_i)$. We denote the $n$-fold Cartesian product of $\mathcal{M}$ with itself as $\mathcal{M}^n = \mathcal{M} \times \cdots \times \mathcal{M}$, and use $[n] := \{1, 2, \cdots, n\}$. For any $x \in \mathcal{M}$, we denote the tangent space and normal space of $\mathcal{M}$ at $x$ as $T_x\mathcal{M}$ and $N_x\mathcal{M}$, respectively. We mark $\|\cdot\|$ as the Euclidean norm. The Euclidean gradient of $f$ is $\nabla f(x)$ and the Riemannian gradient of $f$ is $\operatorname{grad} f(x)$. Let $I_d$ and $\mathbf{1}_n \in \mathbb{R}^n$ be the $d \times d$ identity matrix and a vector of all entries one, respectively. Let $\mathbf{W}^t := W^t \otimes I_d$, where $t$ is a positive integer and $\otimes$ denotes the Kronecker product.

## 2.2 RIEMANNIAN MANIFOLD

We define the distance of a point $x \in \mathbb{R}^{d \times r}$ onto $\mathcal{M}$ by

$$\operatorname{dist}(x, \mathcal{M}) := \inf_{y \in \mathcal{M}} \|y - x\|,$$

then, for any radius $R > 0$, the $R$-tube around $\mathcal{M}$ can be defined as the set:

$$U_\mathcal{M}(R) := \{x : \operatorname{dist}(x, \mathcal{M}) \le R\}.$$

Furthermore, we define the nearest-point projection of a point $x \in \mathbb{R}^{d \times r}$ onto $\mathcal{M}$ by

$$\mathcal{P}_\mathcal{M}(x) := \arg\min_{y \in \mathcal{M}} \|y - x\|.$$

Based on Definition 1, it should be noted that a closed set $\mathcal{M}$ is $R$-proximally smooth if the projection $\mathcal{P}_\mathcal{M}(x)$ is a singleton whenever $\operatorname{dist}(x, \mathcal{M}) < R$. In particular, when $\mathcal{M}$ is the Stiefel manifold, it is a 1-proximally smooth set (Balashov & Kamalov, 2021). And these properties will be crucial for us to demonstrate the convergence.

**Definition 1** *Clarke et al. (1995) An R-**proximally smooth** set $\mathcal{M}$ satisfies that for any real $\gamma \in (0, R)$, the estimate holds:*

$$\|\mathcal{P}_\mathcal{M}(x) - \mathcal{P}_\mathcal{M}(y)\| \le \frac{R}{R - \gamma} \|x - y\|, \quad \forall x, y \in U_\mathcal{M}(\gamma). \tag{2}$$

To proceed the optimization on Riemannian manifolds, we introduce a key concept called the retraction operator in Definition 2. Obviously, the exponential maps (Absil et al., 2008) also satisfies this definition, so that the retraction operator is not unique.

**Definition 2** *Absil et al. (2008) A smooth map $\mathcal{R} : T\mathcal{M} \to \mathcal{M}$ is called a **retraction** on a smooth manifold $\mathcal{M}$ if the retraction of $\mathcal{R}$ to the tangent space $T_x\mathcal{M}$ at any point $x \in \mathcal{M}$, denoted by $\mathcal{R}_x$, satisfies the following conditions:*
*(i) $\mathcal{R}$ is continuously differentiable.*
*(ii) $\mathcal{R}_x(0_x) = x$, where $0_x$ is the zero element of $T_x\mathcal{M}$.*
*(iii) $D\mathcal{R}_x(0_x) = \operatorname{id}_{T_x\mathcal{M}}$, the identity mapping on $T_x\mathcal{M}$.*

Furthermore, we can introduce a well-known concept called a vector transport, which as a special case of parallel translation can be explicitly formulated on the Stiefel manifold. Compared to parallel translation, a vector transport is easier and cheaper to compute (Sato, 2021). Using the Whitney sum $T\mathcal{M} \oplus T\mathcal{M} := \{(\eta, \xi) | \eta, \xi \in T_x\mathcal{M}, x \in \mathcal{M}\}$, we can define a vector transport as follows.

**Definition 3** *Absil et al. (2008) A map $\mathcal{T} : T\mathcal{M} \oplus T\mathcal{M} \to T\mathcal{M} : (\eta, \xi) \mapsto \mathcal{T}_\eta(\xi)$ is called a **vector transport** on $\mathcal{M}$ if there exists a retraction $\mathcal{R}$ on $\mathcal{M}$ and $\mathcal{T}$ satisfies the following conditions for any $x \in \mathcal{M}$:*
*(i) $\mathcal{T}_\eta(\xi) \in T_{\mathcal{R}_x(\eta)}\mathcal{M}, \quad \eta, \xi \in T_x\mathcal{M}.$*
*(ii) $\mathcal{T}_{0_x}(\xi) = \xi, \quad \xi \in T_x\mathcal{M}.$*
*(iii) $\mathcal{T}_\eta(a\xi + b\zeta) = a\mathcal{T}_\eta(\xi) + b\mathcal{T}_\eta(\zeta), \quad a, b \in \mathbb{R}, \quad \eta, \xi, \zeta \in T_x\mathcal{M}.$*

**Example 1** *Absil et al. (2008) On a Riemannian manifold $\mathcal{M}$ with a retraction $\mathcal{R}$, we can construct a vector transport $\mathcal{T}^R : T\mathcal{M} \oplus T\mathcal{M} \to T\mathcal{M} : (\eta, \xi) \mapsto \mathcal{T}_\eta^R(\xi)$ defined by*

$$\mathcal{T}_\eta^R(\xi) := D\mathcal{R}_x(\eta)[\xi], \quad \eta, \xi \in T_x\mathcal{M}, \quad x \in \mathcal{M},$$

*called the differentiated retraction.*

## 3   Consensus problem on Stiefel manifold

Let $x_1, \cdots, x_n \in \mathcal{M}$ be the local variables of each agent, we denote the Euclidean average point of $x_1, \cdots, x_n$ by

$$\hat{x} := \frac{1}{n} \sum_{i=1}^{n} x_i. \tag{3}$$

In Euclidean space, one can use $\sum_{i=1}^{n} \|x_i - \hat{x}\|^2$ to measure the consensus error. Instead, on the Stiefel manifold $\mathrm{St}(d, r)$, we use the induced arithmetic mean (Sarlette & Sepulchre, 2009), defined as follows:

$$\bar{x} := \arg \min_{y \in \mathrm{St}(d,r)} \sum_{i=1}^{n} \|y - x_i\|^2 = \mathcal{P}_{\mathrm{St}}(\hat{x}), \tag{4}$$

where $\mathcal{P}_{\mathrm{St}}(\cdot)$ is the orthogonal projection onto $\mathrm{St}(d, r)$. Considering the Riemannian optimization, the Riemannian gradient of $f_i(x)$ on $\mathrm{St}(d, r)$, endowed with the induced Riemannian metric from the Euclidean inner product $\langle \cdot, \cdot \rangle$, is given by

$$\mathrm{grad}\, f_i(x) = \mathcal{P}_{T_x \mathcal{M}}(\nabla f_i(x)), \tag{5}$$

where $\mathcal{P}_{T_x \mathcal{M}}(\cdot)$ is the orthogonal projection onto $T_x \mathcal{M}$. More specifically (Edelman et al., 1998; Absil et al., 2008), for any $y \in \mathbb{R}^{d \times r}$, we have

$$\mathcal{P}_{T_x \mathcal{M}}(y) = y - \frac{1}{2} x (x^\top y + y^\top x). \tag{6}$$

Subsequently, the $\epsilon$-stationary point of problem (1) is given by Definition 4.

**Definition 4** *Chen et al. (2021) The set of points* $\mathbf{x}^\top = (x_1^\top x_2^\top \cdots x_n^\top)$ *is called an $\epsilon$-stationary point of problem (1) if the following holds:*

$$\frac{1}{n} \sum_{i=1}^{n} \|x_i - \bar{x}\|^2 \leq \epsilon \quad \text{and} \quad \|\mathrm{grad}\, f(\bar{x})\|^2 \leq \epsilon, \tag{7}$$

*where* $f(\bar{x}) = \frac{1}{n} \sum_{i=1}^{n} f_i(\bar{x})$.

To achieve the stationary point given in Definition 4, the consensus problem over $\mathrm{St}(d, r)$ needs to be considered to minimize the following quadratic loss function

$$\min \varphi^t(\mathbf{x}) := \frac{1}{4} \sum_{i=1}^{n} \sum_{j=1}^{n} W_{ij}^t \|x_i - x_j\|^2,$$
$$\text{s.t. } x_i \in \mathcal{M}, \forall i \in [n], \tag{8}$$

where the positive integer $t$ is used to indicate the $t$-th power of the doubly stochastic matrix $W$. Note that $W_{ij}^t$ is computed through performing $t$ steps of communication on the tangent space, and satisfies the following assumption.

**Assumption 1** *We assume that the undirected graph $G$ is connected and $W$ is doubly stochastic, i.e., (i) $W = W^\top$; (ii) $W_{ij} \geq 0$ and $0 < W_{ii} < 1$ for all $i, j$; (iii) Eigenvalues of $W$ lie in $(-1, 1]$. The second largest singular value $\sigma_2$ of $W$ lies in $[0, 1)$.*

Throughout the paper, we assume that the local function $f_i(x)$ is Lipschitz smooth, which is a standard assumption in theoretical analysis of the optimization problem (Jorge & Stephen, 2006; Zeng & Yin, 2018; Deng & Hu, 2023).

**Assumption 2** *Each local function $f_i(x)$ has L-Lipschitz continuous gradient*

$$\|\nabla f_i(x) - \nabla f_i(y)\| \leq L \|x - y\|, \quad i \in [n], \tag{9}$$

*and let $L_f := \max_{x \in \mathrm{St}(d,r)} \|\nabla f_i(x)\|$. Therefore, $\nabla f(x)$ is also L-Lipschitz continuous in the Euclidean space and $L_f \geq \max_{x \in \mathrm{St}(d,r)} \|\nabla f(x)\|$.*

With the properties of projection operators, we can derive the similar Lipschitz inequality on the Stiefel manifold as the Euclidean-type one (Nesterov, 2013) in the following lemma.

**Lemma 1** *(Lipschitz-type inequality) Under Assumption 2, for any $x, y \in \text{St}(d, r)$, if $f(x)$ is L-Lipschitz smooth in the Euclidean space, then there exists a constant $L_g = L + 2L_f$ such that*

$$\| \operatorname{grad} f_i(x) - \operatorname{grad} f_i(y) \| \leq L_g \|x - y\|, \quad i \in [n]. \tag{10}$$

*Proof.* The proofs can be found in Appendix A. $\qquad\square$

Furthermore, since the Stiefel manifold is a 1-proximally smooth set (Balashov & Kamalov, 2021), the projection operator on $\text{St}(d, r)$ has the following property based on Definition 1

$$\|\mathcal{P}_{\mathcal{M}}(x) - \mathcal{P}_{\mathcal{M}}(y)\| \leq \frac{1}{1 - \gamma} \|x - y\|, \quad \forall x, y \in U_{\mathcal{M}}(\gamma), \gamma \in (0, 1). \tag{11}$$

This inequality will be used to characterize the local convergence of the consensus problem.

## 4 Decentralized Riemannian conjugate gradient method

In this section, we will present a decentralized Riemannian conjugate gradient descent (DRCGD) method for solving the problem (1) described in Algorithm 1 and yield the convergence analysis.

### 4.1 The algorithm

We now introduce conjugate gradient methods on a Riemannian manifold $\mathcal{M}$. Our goal is to develop the decentralized version of Riemannian conjugate gradient methods on $\text{St}(d, r)$. The generalized Riemannian conjugate gradient descent (Absil et al., 2008; Sato, 2021) iterates as

$$x_{k+1} = \mathcal{R}_{x_k}(\alpha_k \eta_k), \tag{12}$$

where $\eta_k$ is the search direction on the tangent space $T_{x_k}\mathcal{M}$ and $\alpha_k > 0$ is the step size. Then an operation called retraction $\mathcal{R}_{x_k}$ is performed to ensure feasibility, whose definition is given in Definition 2. It follows from Definition 3 that we have $\mathscr{T}_{\alpha_k \eta_k}(\eta_k) \in T_{x_{k+1}}\mathcal{M}$. Thus, the search direction (Sato, 2021) can be iterated as

$$\eta_{k+1} = -\operatorname{grad} f(x_{k+1}) + \beta_{k+1} \mathscr{T}_{\alpha_k \eta_k}(\eta_k), \quad k = 0, 1, \cdots, \tag{13}$$

where the scalar $\beta_{k+1} \in \mathbb{R}$. Since $\operatorname{grad} f(x_{k+1}) \in T_{x_{k+1}}\mathcal{M}$ and $\beta_{k+1}\eta_k \in T_{x_k}\mathcal{M}$, they belong to different tangent spaces and cannot be added. Hence, the vector transport in Definition 3 needs to be used to map a tangent vector in $T_{x_k}\mathcal{M}$ to one in $T_{x_{k+1}}\mathcal{M}$.

However, vector transports are still computationally expensive, which significantly affects the efficiency of our algorithm. To extend search directions in the decentralized scenario together with computationally cheap needs, we perform the following update of decentralized search directions:

$$\eta_{i,k+1} = -\operatorname{grad} f_i(x_{i,k+1}) + \beta_{i,k+1} \mathcal{P}_{T_{x_{i,k+1}}\mathcal{M}} \left( \sum_{j=1}^{n} W_{ij}^t \eta_{j,k} \right), \quad i \in [n], \tag{14}$$

where $\operatorname{grad} f_i(x_{i,k+1}) \in T_{x_{i,k+1}}\mathcal{M}$ and $\eta_{i,k} \in T_{x_{i,k}}\mathcal{M}$. Note that $\sum_{j=1}^{n} W_{ij}^t \eta_{j,k}$ is clearly not on the tangent space $T_{x_{i,k+1}}\mathcal{M}$ and even not on the tangent space $T_{x_{i,k}}\mathcal{M}$. Therefore, it is important to define the projection $\mathcal{P}_{T_{x_{i,k+1}}\mathcal{M}}$ of $\sum_{j=1}^{n} W_{ij}^t \eta_{j,k}$ to $T_{x_{i,k+1}}\mathcal{M}$ so that we can compute the addition in the same tangent space $T_{x_{i,k+1}}\mathcal{M}$ to update the $\eta_{i,k+1}$. Simultaneously, $\sum_{j=1}^{n} W_{ij}^t \eta_{j,k}$ also achieves the consensus of search directions. On the other hand, similar to the decentralized projected Riemannian gradient descent (Deng & Hu, 2023), the DRCGD performs the following update in the $k$-th iteration

$$x_{i,k+1} = \mathcal{P}_{\mathcal{M}} \left( \sum_{j=1}^{n} W_{ij}^t x_{j,k} + \alpha_k \eta_{i,k} \right), \quad i \in [n]. \tag{15}$$

The Riemanian gradient step with a unit step size, i.e., $\mathcal{P}_{\mathcal{M}} \left( \sum_{j=1}^{n} W_{ij}^t x_{j,k} \right)$, is utilized in the above iteration for the consensus problem (8). So far, we have presented the efficient method by replacing both retractions and vector transports with projection operators.

Regarding $\beta_{k+1}$, there are six standard types in the Euclidean space, which were proposed by Fletcher & Reeves (1964), Dai & Yuan (1999), Fletcher (2000), Polak & Ribiere (1969) and Polyak (1969), Hestenes et al. (1952), and Liu & Storey (1991), respectively. Furthermore, the Riemannian version of $\beta_{k+1}$ was given in (Sato, 2022). Ring & Wirth (2012) analyzed the Riemannian conjugate gradient with a specific scalar $\beta_{k+1}^{\mathrm{R-FR}}$, which is a natural generalization of $\beta_{k+1}^{\mathrm{FR}}$ in (Fletcher & Reeves, 1964). In this paper, we yield a naive extension of $\beta_{k+1}^{\mathrm{R-FR}}$ in terms of the decentralized type

$$\beta_{i,k+1}^{\mathrm{R-FR}} = \frac{\langle \operatorname{grad} f_i(x_{i,k+1}), \operatorname{grad} f_i(x_{i,k+1}) \rangle_{x_{i,k+1}}}{\langle \operatorname{grad} f_i(x_{i,k}), \operatorname{grad} f_i(x_{i,k}) \rangle_{x_{i,k}}} = \frac{\|\operatorname{grad} f_i(x_{i,k+1})\|_{x_{i,k+1}}^2}{\|\operatorname{grad} f_i(x_{i,k})\|_{x_{i,k}}^2}, \qquad (16)$$

where the "R−" stands for "Riemannian" and "FR" stands for "Fletcher-Reeves" type (Fletcher & Reeves, 1964). With the above preparations, we present the DRCGD method described in Algorithm 1. The step 3 first performs a consensus step and then updates the local variable using search directions $\eta_{i,k}$. The step 4 uses the decentralized version of $\beta_{k+1}^{\mathrm{R-FR}}$. The step 5 is to project the search direction onto the tangent space $T_{x_{i,k+1}}\mathcal{M}$, which follows a projection update.

---

**Algorithm 1** Decentralized Riemannian Conjugate Gradient Descent (DRCGD) for solving Eq.(1).

**Input:** Initial point $\mathbf{x}_0 \in \mathcal{M}^n$, an integer $t$, set $\eta_{i,0} = -\operatorname{grad} f_i(x_{i,0})$.
1: **for** $k = 0, \cdots$ **do**                                            ▷ for each node $i \in [n]$, in parallel
2:     Choose diminishing step size $\alpha_k = \mathcal{O}(1/\sqrt{k})$
3:     Update $x_{i,k+1} = \mathcal{P}_{\mathcal{M}} \left( \sum_{j=1}^n W_{ij}^t x_{j,k} + \alpha_k \eta_{i,k} \right)$
4:     Compute $\beta_{i,k+1} = \| \operatorname{grad} f_i(x_{i,k+1})\|^2 / \| \operatorname{grad} f_i(x_{i,k})\|^2$
5:     Update $\eta_{i,k+1} = -\operatorname{grad} f_i(x_{i,k+1}) + \beta_{i,k+1}\mathcal{P}_{T_{x_{i,k+1}}\mathcal{M}} \left( \sum_{j=1}^n W_{ij}^t \eta_{j,k} \right)$
6: **end for**

---

To analyze the convergence of the proposed algorithm, the following assumptions on the step size $\alpha_k$ are also needed (Sato, 2022).

**Assumption 3** *The step size $\alpha_k > 0$ satisfies the following conditions:*
*(i) $\alpha_k$ is decreasing and bounded*

$$\lim_{k \to \infty} \alpha_k = 0, \quad \lim_{k \to \infty} \frac{\alpha_{k+1}}{\alpha_k} = 1, \quad 0 < \alpha_k \leq \frac{\gamma(1-\gamma)}{4C}. \qquad (17)$$

*(ii) For constant $c_1$ and $c_2$ with $0 < c_1 < c_2 < 1$, the Armijo condition on $\mathcal{M}$ is*

$$f_i(x_{i,k+1}) \leq f_i(x_{i,k}) + c_1\alpha_k\langle\operatorname{grad} f_i(x_{i,k}), \eta_{i,k}\rangle_{x_{i,k}}. \qquad (18)$$

*(iii) The strong Wolfe condition is*

$$\left| \left\langle \operatorname{grad} f_i\left(x_{i,k+1}\right), \mathscr{T}_{\alpha_k\eta_{i,k}}^R(\eta_{i,k}) \right\rangle_{x_{i,k+1}} \right| \leq c_2 \left| \langle \operatorname{grad} f_i\left(x_{i,k}\right), \eta_{i,k}\rangle_{x_{i,k}} \right|. \qquad (19)$$

## 4.2 CONVERGENCE ANALYSIS

This subsection focuses on the global convergence analysis of our DRCGD algorithm. Different from the bounded assumption of a vector transport in (Sato, 2022), we give an extended assumption about the projection operator in the decentralized scenario, where $g_{i,k+1} := \operatorname{grad} f_i(x_{i,k+1})$. Specifically, we assume, for each $k \geq 0$, that the following inequality holds

$$\left| \left\langle g_{i,k+1}, \mathcal{P}_{T_{x_{i,k+1}}\mathcal{M}} \left( \sum_{j=1}^n W_{ij}^t \eta_{j,k} \right) \right\rangle_{x_{i,k+1}} \right| \leq \left| \left\langle g_{i,k+1}, \mathscr{T}_{\alpha_k\eta_{i,k}}^R(\eta_{i,k}) \right\rangle_{x_{i,k+1}} \right|, \qquad (20)$$

which will be used as a substitute to proceed the following demonstration. Next, we consider proving the convergence of the Fletcher-Reeves-type DRCGD method for each agent, i.e., we use $\beta_{i,k+1}^{\mathrm{R-FR}}$ in Eq.(16). See Al-Baali (1985) for its Euclidean version and Sato (2022) for its Riemannian version.

At last, we establish the global convergence based on Theorem 2 and Theorem 3, which give the locally linear convergence of consensus error and the convergence of each agent, respectively.

**Theorem 1** (*Global convergence*). *Let* $\{\mathbf{x}_k\}$ *be the sequence generated by Algorithm 1. Suppose that Assumptions 1 and 2 hold. If* $\mathbf{x}_0 \in \mathcal{N} := \{\mathbf{x} : \|\hat{x} - \bar{x}\| \leq \gamma/2\}$ *and* $\|\beta_{i,k}\| \leq C$ *(a constant* $C > 0$*), then*

$$\lim_{k \to \infty} \inf \| \operatorname{grad} f(\bar{x}_k)\|^2 = 0. \tag{21}$$

*Proof.* The proofs can be found in Appendix D.1. $\qquad\square$

## 5 NUMERICAL EXPERIMENT

In this section, we compare our DRCGD method with DRDGD (Chen et al., 2021) and DPRGD (Deng & Hu, 2023), which are first-order decentralized Riemannian optimization methods using retraction and projection respectively, on the following decentralized eigenvector problem:

$$\min_{\mathbf{x} \in \mathcal{M}^n} -\frac{1}{2n} \sum_{i=1}^n \operatorname{tr}\left(x_i^\top A_i^\top A_i x_i\right), \quad \text{s.t.} \quad x_1 = \ldots = x_n, \tag{22}$$

where $\mathcal{M}^n := \underbrace{\operatorname{St}(d, r) \times \cdots \times \operatorname{St}(d, r)}_{n}$, $A_i \in \mathbb{R}^{m_i \times d}$ is the local data matrix for agent $i$ and $m_i$ is the sample size. Note that $A^\top := [A_1^\top, A_2^\top, \cdots, A_n^\top]$ is the global data matrix. For any solution $x^*$ of Eq.(22), giving an orthogonal matrix $Q \in \mathbb{R}^{r \times r}$, $x^* Q$ is also a solution in essence. Then the distance between two points $x$ and $x^*$ can be defined as

$$d_s(x, x^*) = \min_{Q^\top Q = QQ^\top = I_r} \|xQ - x^*\|.$$

We employ fixed step sizes for all comparisons, i.e., the step size is set to $\alpha_k = \frac{\hat{\alpha}}{\sqrt{K}}$ with $K$ being the maximal number of iterations. We examine various graph matrices used to represent the topology across agents, i.e., the Erdos-Renyi (ER) network with probability $p$ and the Ring network. It follows from (Chen et al., 2021) that $W$ is the Metroplis constant matrix (Shi et al., 2015).

We measure algorithms by four metrics, i.e., the consensus error $\|\mathbf{x}_k - \bar{\mathbf{x}}_k\|$, the gradient norm $\|\operatorname{grad} f(\bar{x}_k)\|$, the objective function $f(\bar{x}_k) - f^*$, and the distance to the global optimum $d_s(\bar{x}_k, x^*)$, respectively. The experiments are evaluated with the Intel(R) Core(TM) i7-12700 CPU. And the codes are implemented in Python with mpi4py.

### 5.1 SYNTHETIC DATA

We fix $m_1 = m_2 = \cdots = m_n = 1000$, $d = 10$, and $r = 5$. Then we generate $m_1 \times n$ independent and identically distributed samples to obtain $A$ by following standard multi-variate Gaussian distribution. Specifically, let $A = U\Sigma V^\top$ be the truncated SVD, where $U \in \mathbb{R}^{1000n \times d}$ and $V \in \mathbb{R}^{d \times d}$ are orthogonal matrices, and $\Sigma \in \mathbb{R}^{d \times d}$ is a diagonal matrix. Then we set the singular values of $A$ to be $\Sigma_{i,i} = \Sigma_{0,0} \times \Delta^{i/2}$ where $i \in [d]$ and eigengap $\Delta \in (0, 1)$. We also fix the maximum iteration epoch to 200 and early terminate it if $d_s(\bar{x}_k, x^*) \leq 10^{-5}$.

The comparison results are shown in Figures 1, 2, and 3. It can be seen from Figure 1 that our DRCGD converges faster than DPRGD under different numbers of agents ($n = 16$ and $n = 32$). When $n$ becomes larger, these two algorithms both converge slower. In Figure 2, DRDGD gives very similar performance under different numbers of consensus steps, i.e., $t \in \{1, 10, \infty\}$, which means that the numbers of consensus steps do not affect the performance of DRDGD much. A similar phenomenon can be observed in DPRGD. In contrast, as the communication rounds $t$ increase, our DRCGD consistently achieves better performance. Note that one can achieve the case of $t \to \infty$ through a complete graph with the equally weighted matrix. For Figure 3, we see DPRGD has very close trajectories under different graphs on the four metrics. In fact, this also occurs for DRDGD. However, the connected graph ER helps our DRCGD obtain a better final solution than the connected graph Ring because ER network with the probability of each edge is a better graph connection than Ring network. Moreover, our DRCGD with ER $p = 0.6$ performs better than that with ER $p = 0.3$. In conclusion, DRCGD always converges faster and performs better than both DRDGD and DPRGD under different network graphs because the search direction of conjugate gradient method we designed in Eq.(14) is not only vector transport-free, but also achieves the consensus.

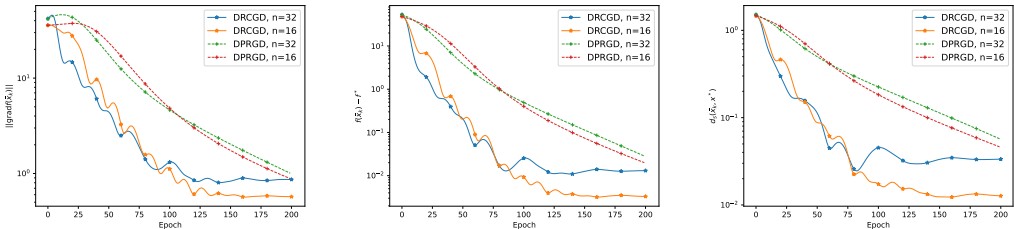

Figure 1: Numerical results on synthetic data with different numbers of agents, eigengap $\Delta = 0.8$, Graph: Ring, $t = 1$, $\hat{\alpha} = 0.01$. y-axis: log-scale.

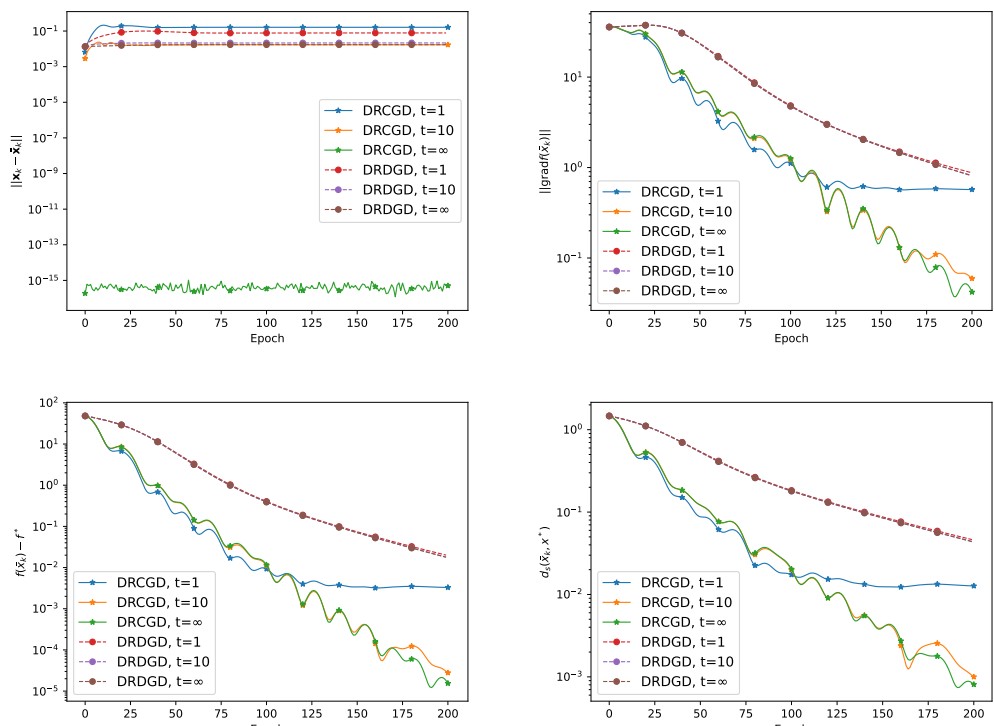

Figure 2: Numerical results on synthetic data with different numbers of consensus steps, eigengap $\Delta = 0.8$, Graph: Ring, $n = 16$, $\hat{\alpha} = 0.01$. y-axis: log-scale.

## 5.2 REAL-WORLD DATA

We also present some numerical results on the MNIST dataset (LeCun, 1998). For MNIST, the samples consist of 60000 hand-written images where the dimension of each image is given by $d = 784$. And these samples make up the data matrix of $60000 \times 784$, which is randomly and evenly partitioned into $n$ agents. We normalize the data matrix by dividing 255. Then each agent holds a local data matrix $A_i$ of $\frac{60000}{n} \times 784$. For brevity, we fix $t = 1$, $r = 5$, and $d = 784$, respectively. $W$ is the Metroplis constant matrix and the graph is the Ring network. The step size of our DRCGD, DRDGD, and DPRGD is $\alpha_k = \frac{\hat{\alpha}}{60000}$. We set the maximum iteration epoch to 1000 and early terminate it if $d_s(\bar{x}_k, x^*) \le 10^{-5}$.

The results for MNIST data with $n = 20$ are shown in Figure 4. We see that the performance of DRDGD and DPRGD are almost the same. When $\hat{\alpha}$ becomes larger, all algorithms converge faster. And our DRCGD converges much faster than both DRDGD and DPRGD.

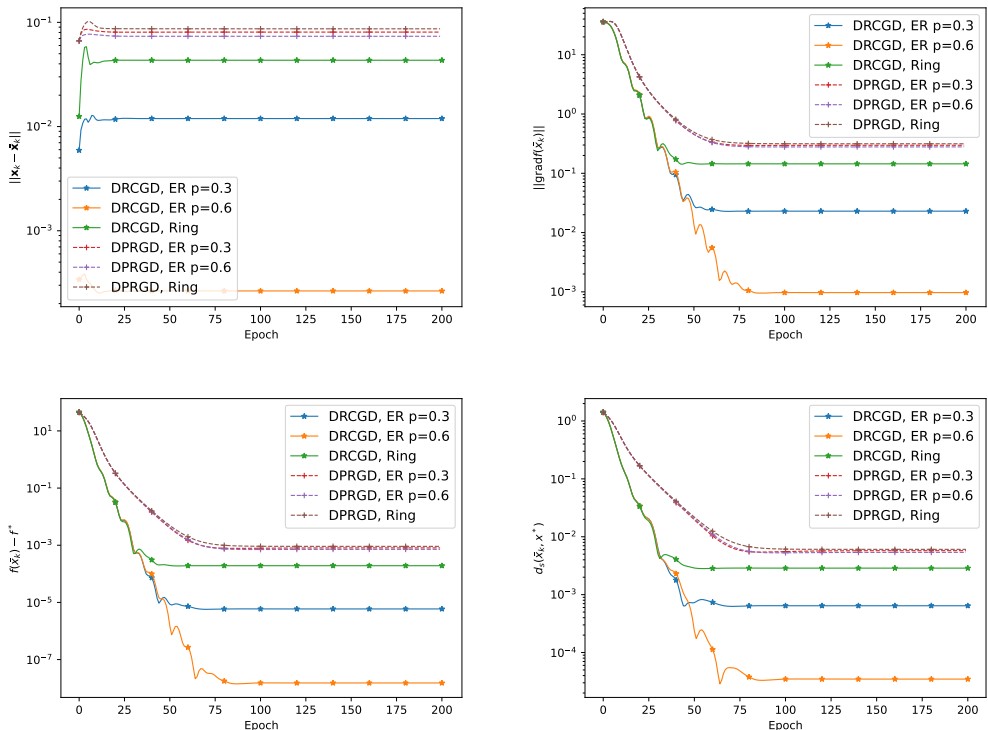

Figure 3: Numerical results on synthetic data with different network graphs, eigengap $\Delta = 0.8$, $t = 10$, $n = 16$, $\hat{\alpha} = 0.05$. y-axis: log-scale.

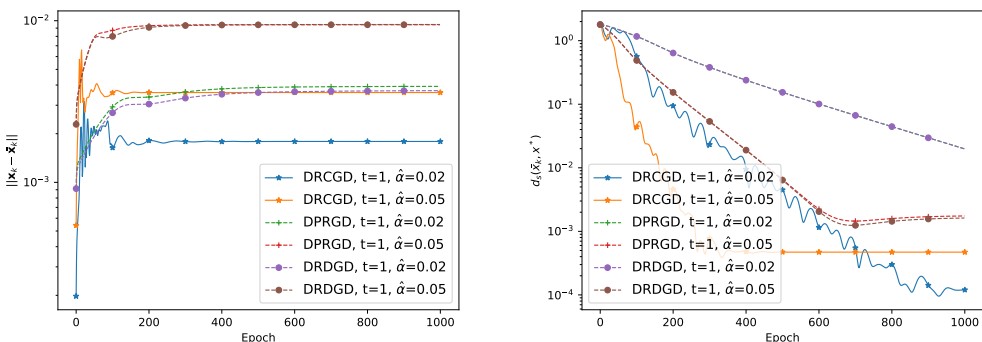

Figure 4: Numerical results on MNIST data with single-step consensus, Graph: Ring, $n = 20$.

## 6 CONCLUSION

We proposed the decentralized Riemannian conjugate gradient method for solving decentralized optimization over the Stiefel manifold. In particular, it is the first decentralized version of the Riemannian conjugate gradient. By replacing retractions and vector transports with projection operators, the global convergence was established under an extended assumption (20) on the basis of (Sato, 2021), thereby reducing the computational complexity required by each agent. Numerical results demonstrated the effectiveness of our proposed algorithm. In the future, we will further extend our algorithm to a compact sub-manifold. On the other hand, it will be interesting to develop the decentralized version of online optimization over Riemannian manifolds.

## ACKNOWLEDGMENTS

This work was supported by NSFC 62088101 Autonomous Intelligent Unmanned Systems.

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

# A    PROOFS FOR LEMMA 1

*Proof.* Since $\text{grad } f_i(x) = \mathcal{P}_{T_x\mathcal{M}}(\nabla f_i(x))$, we have

$$
\begin{aligned}
\|\text{grad } f_i(x) - \text{grad } f_i(y)\| &= \left\|\mathcal{P}_{T_x\mathcal{M}}(\nabla f_i(x)) - \mathcal{P}_{T_y\mathcal{M}}(\nabla f_i(y))\right\| \\
&= \left\|\mathcal{P}_{T_x\mathcal{M}}(\nabla f_i(x)) - \mathcal{P}_{T_x\mathcal{M}}(\nabla f_i(y)) + \mathcal{P}_{T_x\mathcal{M}}(\nabla f_i(y)) - \mathcal{P}_{T_y\mathcal{M}}(\nabla f_i(y))\right\| \\
&\leq \left\|\mathcal{P}_{T_x\mathcal{M}}(\nabla f_i(x) - \nabla f_i(y))\right\| + \left\|\mathcal{P}_{T_x\mathcal{M}}(\nabla f_i(y)) - \mathcal{P}_{T_y\mathcal{M}}(\nabla f_i(y))\right\| \\
&\leq \|\nabla f_i(x) - \nabla f_i(y)\| + \left\|\mathcal{P}_{T_x\mathcal{M}}(\nabla f_i(y)) - \mathcal{P}_{T_y\mathcal{M}}(\nabla f_i(y))\right\| \\
&\leq \|\nabla f_i(x) - \nabla f_i(y)\| + 2L_f\|x - y\| \\
&\leq (L + 2L_f)\|x - y\|,
\end{aligned}
\tag{23}
$$

where, by Eq.(6), the third inequality uses

$$
\begin{aligned}
&\left\|\mathcal{P}_{T_x\mathcal{M}}(\nabla f_i(y)) - \mathcal{P}_{T_y\mathcal{M}}(\nabla f_i(y))\right\| \\
&= \frac{1}{2}\left\|x\left(x^\top \nabla f_i(y) + \nabla f_i(y)^\top x\right) - y\left(y^\top \nabla f_i(y) + \nabla f_i(y)^\top y\right)\right\| \\
&\leq \frac{1}{2}\left(\left\|x\left((x-y)^\top \nabla f_i(y) + \nabla f_i(y)^\top (x-y)\right)\right\| + \left\|(x-y)\left(y^\top \nabla f_i(y) + \nabla f_i(y)^\top y\right)\right\|\right) \\
&\leq \frac{1}{2}\left(2\|x\| \cdot \|x-y\| \cdot \|\nabla f_i(y)\| + 2\|x-y\| \cdot \|\nabla f_i(y)\| \cdot \|y\|\right) \\
&\leq 2\|x-y\| \cdot \|\nabla f_i(y)\| \leq 2\max_{y\in\text{St}(d,r)}\|\nabla f_i(y)\| \cdot \|x-y\| = 2L_f\|x-y\|.
\end{aligned}
\tag{24}
$$

The proof is completed. $\qquad\square$

# B    LINEAR CONVERGENCE OF CONSENSUS ERROR

Let us first present the linear convergence of consensus error. For the iteration scheme $x_{i,k+1} = \mathcal{P}_\mathcal{M}\left(\sum_{j=1}^n W_{ij}^t x_{j,k} + \alpha_k \eta_{i,k}\right)$ where $\alpha_k > 0$ and $\eta_{i,k} \in T_{x_{i,k}}\mathcal{M}$, the following lemma yields that, for $\mathbf{x}_k$ in the neighborhood $\mathcal{N}$, the iterates $\mathbf{x}_{k+1}$ also remain in this neighborhood $\mathcal{N}$.

**Lemma 2** *Let $x_{i,k+1} = \mathcal{P}_\mathcal{M}\left(\sum_{j=1}^n W_{ij}^t x_{j,k} + \alpha_k \eta_{i,k}\right)$. On the basis of **Assumption 1**, if $\mathbf{x}_k \in \mathcal{N} := \{\mathbf{x} : \|\hat{x} - \bar{x}\| \leq \gamma/2\}$, $\|\eta_{i,k}\| \leq C$, $0 < \alpha_k \leq \frac{\gamma(1-\gamma)}{4C}$, and $t \geq \left\lceil \log_{\sigma_2}\left(\frac{\gamma(1-\gamma)}{4\sqrt{n}\zeta}\right)\right\rceil$ with $\zeta := \max_{x,y\in\mathcal{M}}\|x - y\|$, then*

$$
\sum_{j=1}^n W_{ij}^t x_{j,k} + \alpha_k \eta_{i,k} \in U_\mathcal{M}(\gamma), \quad i = 1, \cdots, n,
\tag{25}
$$

$$
\|\hat{x}_{k+1} - \bar{x}_{k+1}\| \leq \frac{1}{2}\gamma.
\tag{26}
$$

*Proof.* Since $\mathbf{x}_k \in \mathcal{N}$, we have

$$
\begin{aligned}
\|\hat{x}_{k+1} - \bar{x}_{k+1}\| &\leq \|\hat{x}_{k+1} - \bar{x}_k\| \leq \frac{1}{n}\sum_{i=1}^n \|x_{i,k+1} - \bar{x}_k\| \\
&= \frac{1}{n}\sum_{i=1}^n \left\|\mathcal{P}_\mathcal{M}\left(\sum_{j=1}^n W_{ij}^t x_{j,k} + \alpha_k \eta_{i,k}\right) - \mathcal{P}_\mathcal{M}(\hat{x}_k)\right\| \\
&\leq \frac{1}{1-\gamma}\left\|\sum_{j=1}^n W_{ij}^t x_{j,k} + \alpha_k \eta_{i,k} - \hat{x}_k\right\| \leq \frac{1}{2}\gamma,
\end{aligned}
$$

where the third inequality uses Eq.(11) and the fourth inequality yields

$$
\begin{aligned}
\left\| \sum_{j=1}^{n} W_{ij}^t x_{j,k} + \alpha_k \eta_{i,k} - \hat{x}_k \right\| &\leq \left\| \sum_{j=1}^{n} W_{ij}^t x_{j,k} - \hat{x}_k \right\| + \| \alpha_k \eta_{i,k} \| \\
&\leq \left\| \sum_{j=1}^{n} \left( W_{ij}^t - \frac{1}{n} \right) (x_{j,k} - \hat{x}_k) \right\| + \alpha_k C \\
&\leq \sum_{j=1}^{n} \left| W_{ij}^t - \frac{1}{n} \right| \| x_{j,k} - \hat{x}_k \| + \alpha_k C \\
&\leq \zeta \max_i \sum_{j=1}^{n} \left| W_{ij}^t - \frac{1}{n} \right| + \alpha_k C \leq \sqrt{n} \sigma_2^t \zeta + \alpha_k C \leq \frac{\gamma(1-\gamma)}{2},
\end{aligned}
$$

where the fourth inequality uses that $\| x_{j,k} - \hat{x}_k \| \leq \frac{1}{n} \sum_{i=1}^{n} \| x_{j,k} - x_{i,k} \| \leq \zeta$ (Deng & Hu, 2023) and the fifth inequality follows from the bound on the total variation distance between any row of $W^t$ and $\frac{1}{n} \mathbf{1}_n$ (Diaconis & Stroock, 1991; Boyd et al., 2004). For any $i \in [n]$, since $\bar{x}_k \in \mathcal{M}$ and $\gamma \in (0,1)$, we have

$$
\begin{aligned}
\left\| \sum_{j=1}^{n} W_{ij}^t x_{j,k} + \alpha_k \eta_{i,k} - \bar{x}_k \right\| &\leq \left\| \sum_{j=1}^{n} W_{ij}^t x_{j,k} + \alpha_k \eta_{i,k} - \hat{x}_k \right\| + \| \hat{x}_k - \bar{x}_k \| \\
&\leq \frac{\gamma(1-\gamma)}{2} + \frac{1}{2}\gamma < \gamma.
\end{aligned}
$$

The proof is completed. □

In Lemma 2, the search direction $\eta_{i,k}$ is required to be bounded. Let $\eta_{i,k} = 0$, then we can consider the convergence of consensus error.

**Theorem 2** (*Linear convergence of consensus error*). *Let* $x_{i,k+1} = \mathcal{P}_{\mathcal{M}} \left( \sum_{j=1}^{n} W_{ij}^t x_{j,k} \right)$. *On the basis of **Assumption 1**, if* $\mathbf{x}_k \in \mathcal{N} := \{ \mathbf{x} : \| \hat{x} - \bar{x} \| \leq \gamma/2 \}$ *and* $t \geq \max \left\{ \lceil \log_{\sigma_2}(1-\gamma) \rceil, \left\lceil \log_{\sigma_2} \left( \frac{\gamma(1-\gamma)}{4\sqrt{n}\zeta} \right) \right\rceil \right\}$, *then the following linear convergence with rate* $\sigma_2^t/(1-\gamma) < 1$ *holds*

$$
\| \mathbf{x}_{k+1} - \bar{\mathbf{x}}_{k+1} \| \leq \frac{\sigma_2^t}{1-\gamma} \| \mathbf{x}_k - \bar{\mathbf{x}}_k \|
$$

*Proof.* According to Lemma 2, $\| \hat{x}_0 - \bar{x}_0 \| \leq \frac{1}{2}\gamma$ and $\mathbf{x}_k \in \mathcal{N}$ for all $k \geq 0$ holds, then it holds that

$$
\sum_{j=1}^{n} W_{ij}^t x_{j,k} \in U_{\mathcal{M}}(\gamma), \quad i = 1, \cdots, n.
$$

Let $\mathcal{P}_{\mathcal{M}^n}(\mathbf{x})^\top = [\mathcal{P}_{\mathcal{M}}(x_1)^\top, \cdots, \mathcal{P}_{\mathcal{M}}(x_n)^\top]$, then with the iteration scheme $x_{i,k+1} = \mathcal{P}_{\mathcal{M}} \left( \sum_{j=1}^{n} W_{ij}^t x_{j,k} \right)$ we have

$$
\begin{aligned}
\| \mathbf{x}_{k+1} - \bar{\mathbf{x}}_{k+1} \| &\leq \| \mathbf{x}_{k+1} - \bar{\mathbf{x}}_k \| = \left\| \mathcal{P}_{\mathcal{M}^n} (\mathbf{W}^t \mathbf{x}_k) - \mathcal{P}_{\mathcal{M}^n} (\hat{\mathbf{x}}_k) \right\| \\
&\leq \frac{1}{1-\gamma} \| \mathbf{W}^t \mathbf{x}_k - \hat{\mathbf{x}}_k \| = \frac{1}{1-\gamma} \| (W^t \otimes I_d) \mathbf{x}_k - \hat{\mathbf{x}}_k \| \\
&= \frac{1}{1-\gamma} \left\| \left( \left( W^t - \frac{1}{n} \mathbf{1}_n \mathbf{1}_n^\top \right) \otimes I_d \right) (\mathbf{x}_k - \hat{\mathbf{x}}_k) \right\| \\
&\leq \frac{1}{1-\gamma} \sigma_2^t \| \mathbf{x}_k - \hat{\mathbf{x}}_k \| \leq \frac{1}{1-\gamma} \sigma_2^t \| \mathbf{x}_k - \bar{\mathbf{x}}_k \|,
\end{aligned} \tag{27}
$$

where the second inequality uses Eq.(11). The proof is completed. □

On the Stiefel manifold, by utilizing the 1-proximally smooth property of projection operators, we establish the locally linear convergence of consensus error with a rate of $\sigma_2^t/(1-\gamma)$, where $t$ can be any positive integer, which is consistent with the cases in the Euclidean space (Nedic et al., 2010; Nedić et al., 2018).

## C CONVERGENCE OF EACH AGENT

**Lemma 3** *In **Algorithm** 1 with $\beta_{i,k+1} = \beta_{i,k+1}^{\mathrm{R-FR}}$ in Eq. (16) and Eq. (20), assume that $\alpha_k$ satisfies the strong Wolfe conditions in Eq.(18) and Eq.(19) with $0 < c_1 < c_2 < 1/2$, for each $k \geq 0$. If $\mathrm{grad}\, f_i(x_{i,k}) \neq 0$ for each $k \geq 0$, then $\eta_{i,k}$ as a descent direction satisfies*

$$-\frac{1}{1-c_2} \leq \frac{\langle \mathrm{grad}\, f_i(x_{i,k}), \eta_{i,k} \rangle_{x_{i,k}}}{\|\mathrm{grad}\, f_i(x_{i,k})\|_{x_{i,k}}^2} \leq -\frac{1-2c_2}{1-c_2}. \tag{28}$$

*Proof.* For ease of notation, we denote $g_{i,k} := \mathrm{grad}\, f_i(x_{i,k})$. When $k = 0$, $\eta_{i,0} = -g_{i,0}$ is the initial condition and we have

$$\frac{\langle g_{i,0}, \eta_{i,0} \rangle_{x_{i,0}}}{\|g_{i,0}\|_{x_{i,0}}^2} = \frac{\langle g_{i,0}, -g_{i,0} \rangle_{x_{i,0}}}{\|g_{i,0}\|_{x_{i,0}}^2} = -1.$$

Hence, Eq. (28) holds. Supposing that $\eta_{i,k}$ is a descent direction satisfying Eq. (28) for some $k$, we will prove that $\eta_{i,k+1}$ is also a descent and satisfies Eq. (28) in which $k$ is replaced with $k+1$. Based on Eq.(14) and Eq.(16), we yield

$$\begin{aligned}
\frac{\langle g_{i,k+1}, \eta_{i,k+1} \rangle_{x_{i,k+1}}}{\|g_{i,k+1}\|_{x_{i,k+1}}^2} &= \frac{\left\langle g_{i,k+1}, -g_{i,k+1} + \beta_{i,k+1} \mathcal{P}_{T_{x_{i,k+1}}} \mathcal{M} \left( \sum_{j=1}^n W_{ij}^t \eta_{j,k} \right) \right\rangle_{x_{i,k+1}}}{\|g_{i,k+1}\|_{x_{i,k+1}}^2} \\
&= -1 + \frac{\left\langle g_{i,k+1}, \mathcal{P}_{T_{x_{i,k+1}}} \mathcal{M} \left( \sum_{j=1}^n W_{ij}^t \eta_{j,k} \right) \right\rangle_{x_{i,k+1}}}{\|g_{i,k}\|_{x_{i,k}}^2}.
\end{aligned} \tag{29}$$

Similar to (Sato, 2022), the assumption in Eq.(20) and the strong Wolfe condition in Eq.(19) yield

$$\left| \left\langle g_{i,k+1}, \mathcal{P}_{T_{x_{i,k+1}}} \mathcal{M} \left( \sum_{j=1}^n W_{ij}^t \eta_{j,k} \right) \right\rangle_{x_{i,k+1}} \right| \leq c_2 \left| \langle g_{i,k}, \eta_{i,k} \rangle_{x_{i,k}} \right| = -c_2 \langle g_{i,k}, \eta_{i,k} \rangle_{x_{i,k}}. \tag{30}$$

It follows from Eq.(29) and Eq.(30) that

$$-1 + c_2 \frac{\langle g_{i,k}, \eta_{i,k} \rangle_{x_{i,k}}}{\|g_{i,k}\|_{x_{i,k}}^2} \leq \frac{\langle g_{i,k+1}, \eta_{k+1} \rangle_{x_{i,k+1}}}{\|g_{i,k+1}\|_{x_{i,k+1}}^2} \leq -1 - c_2 \frac{\langle g_{i,k}, \eta_{i,k} \rangle_{x_{i,k}}}{\|g_{i,k}\|_{x_{i,k}}^2}.$$

From the induction hypothesis in Eq.(28), i.e., $\langle g_{i,k}, \eta_{i,k} \rangle_{x_{i,k}} / \|g_{i,k}\|_{x_{i,k}}^2 \geq -(1-c_2)^{-1}$, and the assumption $c_2 > 0$, we finally obtain the following inequality

$$-\frac{1}{1-c_2} \leq \frac{\langle g_{i,k+1}, \eta_{i,k+1} \rangle_{x_{i,k+1}}}{\|g_{i,k+1}\|_{x_{i,k+1}}^2} \leq -\frac{1-2c_2}{1-c_2},$$

which also implies $\langle g_{i,k+1}, \eta_{i,k+1} \rangle_{x_{i,k+1}} < 0$. The proof is completed. $\square$

Subsequently, we proceed to the convergence property of each agent. The proof below is based on the Riemannian version given in Sato (2022).

**Theorem 3** *In **Algorithm** 1 with $\beta_{i,k+1} = \beta_{i,k+1}^{\mathrm{R-FR}}$ in Eq. (16) and Eq. (20), assume that $\alpha_k$ satisfies the strong Wolfe conditions in Eq.(18) and Eq.(19) with $0 < c_1 < c_2 < 1/2$, for each $k \geq 0$. If $f_i$ is*

*bounded below and is of class $C^1$, and the Riemannian version (Ring & Wirth, 2012; Sato & Iwai, 2015) of Zoutendijk's Theorem (Nocedal & Wright, 1999) holds, then we yield*

$$\lim_{k \to \infty} \inf \| \operatorname{grad} f_i(x_{i,k}) \|_{x_{i,k}} = 0, \quad i = 1, \cdots, n. \tag{31}$$

*Proof.* We denote $g_{i,k} := \operatorname{grad} f_i(x_{i,k})$ again. If $g_{i,k_0} = 0$ holds for some $k_0$, then we have $\beta_{i,k_0} = 0$ and $\eta_{i,k_0} = 0$ from Eq.(16) and Eq.(14), which implies $x_{i,k_0+1} = \mathcal{P}_{\mathcal{M}} \left( \sum_{j=1}^n W_{ij}^t x_{j,k_0} \right)$. Based on Theorem 2, the consensus error converges such that $x_{i,k_0+1} \to x_{i,k_0}$ holds. Thus, we obtain $g_{i,k} = 0$ for all $k \geq k_0$ so that Eq.(31) holds.

We next consider the case in which $g_{i,k} \neq 0$ for all $k \geq 0$. Let $\theta_{i,k}$ be the angle between $-g_{i,k}$ and $\eta_{i,k}$, i.e.,

$$\cos \theta_{i,k} = \frac{\langle -g_{i,k}, \eta_{i,k} \rangle_{x_{i,k}}}{\| -g_{i,k} \|_{x_{i,k}} \| \eta_{i,k} \|_{x_{i,k}}} = - \frac{\langle g_{i,k}, \eta_{i,k} \rangle_{x_{i,k}}}{\| g_{i,k} \|_{x_{i,k}} \| \eta_{i,k} \|_{x_{i,k}}}. \tag{32}$$

It follows from Eq.(32) and Eq.(28) that

$$\cos \theta_{i,k} \geq \frac{1 - 2c_2}{1 - c_2} \frac{\| g_{i,k} \|_{x_{i,k}}}{\| \eta_{i,k} \|_{x_{i,k}}}. \tag{33}$$

Since the search directions are descent directions from Lemma 3, Zoutendijk's Theorem together with Eq.(33) yields

$$\sum_{k=0}^{\infty} \frac{\| g_{i,k} \|_{x_{i,k}}^4}{\| \eta_{i,k} \|_{x_{i,k}}^2} < \infty. \tag{34}$$

Combined Eq.(28) and Eq.(30), we have

$$\left| \left\langle g_{i,k}, \mathcal{P}_{T_{x_{i,k}} \mathcal{M}} \left( \sum_{j=1}^n W_{ij}^t \eta_{j,k-1} \right) \right\rangle_{x_{i,k}} \right| \leq -c_2 \langle g_{i,k-1}, \eta_{i,k-1} \rangle_{x_{i,k-1}} \leq \frac{c_2}{1 - c_2} \| g_{i,k-1} \|_{x_{i,k-1}}^2. \tag{35}$$

Using Eq.(16), Eq.(20), and Eq.(35), we obtain the recurrence inequality for $\| \eta_{i,k} \|_{x_{i,k}}^2$:

$$\| \eta_{i,k} \|_{x_{i,k}}^2$$
$$= \left\| -g_{i,k} + \beta_{i,k} \mathcal{P}_{T_{x_{i,k}} \mathcal{M}} \left( \sum_{j=1}^n W_{ij}^t \eta_{j,k-1} \right) \right\|_{x_{i,k}}^2$$
$$\leq \| g_{i,k} \|_{x_{i,k}}^2 + 2\beta_{i,k} \left| \left\langle g_{i,k}, \mathcal{P}_{T_{x_{i,k}} \mathcal{M}} \left( \sum_{j=1}^n W_{ij}^t \eta_{j,k-1} \right) \right\rangle_{x_{i,k}} \right| + \beta_{i,k}^2 \left\| \mathcal{P}_{T_{x_{i,k}} \mathcal{M}} \left( \sum_{j=1}^n W_{ij}^t \eta_{j,k-1} \right) \right\|_{x_{i,k}}^2$$
$$\leq \| g_{i,k} \|_{x_{i,k}}^2 + \frac{2c_2}{1 - c_2} \beta_{i,k} \| g_{i,k-1} \|_{x_{i,k-1}}^2 + \beta_{i,k}^2 \| \eta_{i,k-1} \|_{x_{i,k-1}}^2$$
$$= \| g_{i,k} \|_{x_{i,k}}^2 + \frac{2c_2}{1 - c_2} \| g_{i,k} \|_{x_{i,k}}^2 + \beta_{i,k}^2 \| \eta_{i,k-1} \|_{x_{i,k-1}}^2$$
$$= c \| g_{i,k} \|_{x_{i,k}}^2 + \beta_{i,k}^2 \| \eta_{i,k-1} \|_{x_{i,k-1}}^2, \tag{36}$$

where $c := (1 + c_2)/(1 - c_2) > 1$. Note that we assume in the second inequality, for each $k \geq 1$, that $\left\| \sum_{j=1}^n W_{ij}^t \eta_{j,k-1} \right\|_{x_{i,k}} \leq \| \eta_{i,k-1} \|_{x_{i,k-1}}$ holds, which is similar to Formula (4.28) in (Sato,

2021). We can successively use Eq.(36) with Eq.(16) as

$$\|\eta_{i,k}\|^2_{x_{i,k}}$$

$$\leq c \left( \|g_{i,k}\|^2_{x_{i,k}} + \beta^2_{i,k} \|g_{i,k-1}\|^2_{x_{i,k-1}} + \cdots + \beta^2_{i,k}\beta^2_{i,k-1}\cdots\beta^2_{i,2} \|g_{i,1}\|^2_{x_{i,1}} \right) + \beta^2_{i,k}\beta^2_{i,k-1}\cdots\beta^2_{i,1} \|\eta_{i,0}\|^2_{x_{i,0}}$$

$$= c \|g_{i,k}\|^4_{x_{i,k}} \left( \|g_{i,k}\|^{-2}_{x_{i,k}} + \|g_{i,k-1}\|^{-2}_{x_{i,k-1}} + \cdots + \|g_{i,1}\|^{-2}_{x_{i,1}} \right) + \|g_{i,k}\|^4_{x_{i,k}} \|g_{i,0}\|^{-2}_{x_{i,0}}$$

$$< c \|g_{i,k}\|^4_{x_{i,k}} \sum_{j=0}^{k} \|g_{i,j}\|^{-2}_{x_{i,j}} .$$

(37)

We can prove Eq.(31) by contradiction. We first assume that Eq.(31) does not hold. Then there exists a constant $C > 0$ such that $\|g_{i,k}\|_{x_{i,k}} \geq C > 0$ for all $k \geq 0$ because we also assume $g_{i,k} \neq 0$ for all $k \geq 0$ at the same time. Consequently, we have $\sum_{j=0}^{k} \|g_{i,j}\|^2_{x_{i,j}} \leq C^{-2}(k+1)$. Hence, based on Eq.(37), the left hand side of Eq.(34) is evaluated as

$$\sum_{k=0}^{\infty} \frac{\|g_{i,k}\|^4_{x_{i,k}}}{\|\eta_{i,k}\|^2_{x_{i,k}}} \geq \sum_{k=0}^{\infty} \frac{\epsilon^2}{c} \frac{1}{k+1} = \infty,$$

which contradicts Eq.(34). The proof is completed. □

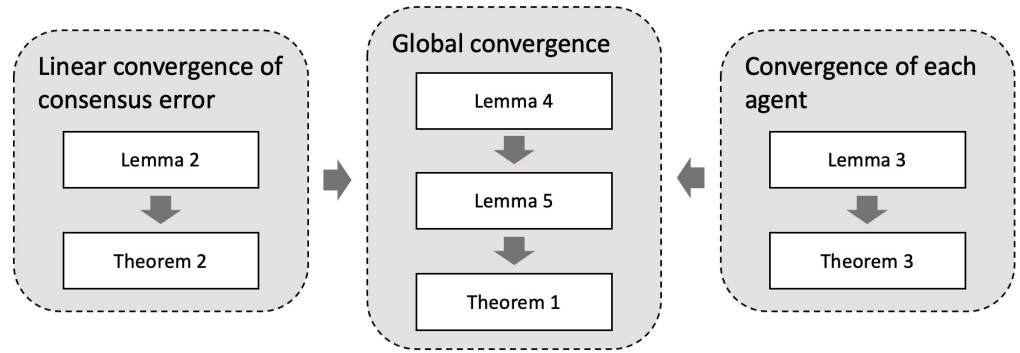

Figure 5: An overview of the proofs.

## D    GLOBAL CONVERGENCE

An overview of this paper is shown in the Figure 5. We now investigate the uniform boundedness of $\|\boldsymbol{\eta}_k\|$ in the following lemma.

**Lemma 4** *Let $\{\mathbf{x}_k\}$ be the sequence generated by Algorithm 1. Suppose that Assumptions 1 and 2 hold. If $\mathbf{x}_0 \in \mathcal{N}$, $\|\beta_{i,k}\| \leq C$, $0 < \alpha_k < \min\left\{\frac{1-\gamma}{8L_g}, \frac{\gamma(1-\gamma)}{4C}\right\}$, and $t \geq \max\left\{\left\lceil\log_{\sigma_2}\left(\frac{1-\gamma}{2\sqrt{n}\zeta}\right)\right\rceil, \left\lceil\log_{\sigma_2}\left(\frac{1}{8\sqrt{n}C}\right)\right\rceil, \left\lceil\log_{\sigma_2}\left(\frac{\gamma(1-\gamma)}{4\sqrt{n}\zeta}\right)\right\rceil\right\}$, it follows that for all $k$, $\mathbf{x}_k \in \mathcal{N}$ and*

$$\|\eta_{i,k}\| \leq 2L_g, \quad \forall i \in [n].$$

(38)

*Proof.* We prove it by induction on both $\|\eta_{i,k}\|$ and $\|\hat{x}_k - \bar{x}_k\|$. Based on Assumption 2, we have $\|\operatorname{grad} f_i(x_{i,k})\| \leq \|\nabla f_i(x_{i,k})\| \leq L_f \leq L_g$ due to $L_g = L + 2L_f \geq L_f$. Then we have $\|\eta_{i,0}\| = \|\operatorname{grad} f_i(x_{i,0})\| \leq L_g$ for all $i \in [n]$ and $\|\hat{x}_0 - \bar{x}_0\| \leq \frac{1}{2}\gamma$. Suppose for some $k \geq 0$

that $\|\eta_{i,k}\| \leq 2L_g$ and $\|\hat{x}_k - \bar{x}_k\| \leq \frac{1}{2}\gamma$. Since $\|\eta_{i,k}\| \leq 2L_g$ and $\alpha_k < \frac{\gamma(1-\gamma)}{4C}$, it follows from Lemma 2 that

$$\sum_{j=1}^{n} W_{ij}^t x_{j,k} + \alpha_k \eta_{i,k} \in U_{\mathcal{M}}(\gamma), \quad i = 1, \cdots, n, \quad \|\hat{x}_{k+1} - \bar{x}_{k+1}\| \leq \frac{1}{2}\gamma.$$

Then, we have

$$\|\eta_{i,k+1} + \operatorname{grad} f_i(x_{i,k})\|$$

$$= \left\| \beta_{i,k+1} \mathcal{P}_{T_{x_{i,k+1}}\mathcal{M}} \left( \sum_{j=1}^{n} W_{ij}^t \eta_{j,k} \right) - (\operatorname{grad} f_i(x_{i,k+1}) - \operatorname{grad} f_i(x_{i,k})) \right\|$$

$$\leq \beta_{i,k+1} \left\| \mathcal{P}_{T_{x_{i,k+1}}\mathcal{M}} \left( \sum_{j=1}^{n} W_{ij}^t \eta_{j,k} \right) \right\| + \|\operatorname{grad} f_i(x_{i,k+1}) - \operatorname{grad} f_i(x_{i,k})\|$$

$$\leq C \left\| \sum_{j=1}^{n} \left( W_{ij}^t - \frac{1}{n} \right) \eta_{j,k} \right\| + L_g \|x_{i,k+1} - x_{i,k}\| \tag{39}$$

$$\leq C\sigma_2^t \sqrt{n} \max_i \|\eta_{i,k}\| + L_g \|x_{i,k+1} - x_{i,k}\|$$

$$\leq C\sigma_2^t \sqrt{n} \max_i \|\eta_{i,k}\| + \frac{L_g}{1-\gamma} \left\| \sum_{j=1}^{n} \left( W_{ij}^t - \frac{1}{n} \right) x_{i,k} + \alpha_k \eta_{i,k} \right\|$$

$$\leq \left( C\sigma_2^t \sqrt{n} + \alpha_k \frac{L_g}{1-\gamma} \right) \max_i \|\eta_{i,k}\| + \frac{L_g}{1-\gamma} \sigma_2^t \sqrt{n}\zeta$$

$$\leq \frac{1}{4} \max_i \|\eta_{i,k}\| + \frac{1}{2}L_g \leq \frac{1}{2}L_g + \frac{1}{2}L_g \leq L_g,$$

where the second inequality uses Lemma 1 and the fourth inequality utilizes Eq.(11). Hence, $\|\eta_{i,k+1}\| \leq \|\eta_{i,k+1} + \operatorname{grad} f_i(x_{i,k})\| + \|\operatorname{grad} f_i(x_{i,k})\| \leq 2L_g$. The proof is completed. □

With the above lemma, we can elaborate the relationship between the consensus error and step size.

**Lemma 5** *Let $\{\mathbf{x}_k\}$ be the sequence generated by Algorithm 1. Suppose that Assumptions 1 and 2 hold. If $\mathbf{x}_0 \in \mathcal{N}$, $\|\eta_{i,k}\| \leq 2L_g$, $t \geq \lceil \log_{\sigma_2}(1-\gamma) \rceil$, and $0 < \alpha_k \leq \frac{\gamma(1-\gamma)}{8L_g}$, it follows that for all $k$, $\mathbf{x}_k \in \mathcal{N}$ and*

$$\frac{1}{n}\|\bar{\mathbf{x}}_k - \mathbf{x}_k\|^2 \leq C\frac{1}{(1-\gamma)^2}L_g^2\alpha_k^2. \tag{40}$$

*Proof.* Since $\|\operatorname{grad} f_i(x_{i,k})\| \leq L_g$ and $\|\hat{x}_0 - \bar{x}_0\| \leq \frac{1}{2}\gamma$, it follows from Lemma 2 that for any $k > 0$, we have

$$\sum_{j=1}^{n} W_{ij}^t x_{j,k} + \alpha_k \eta_{i,k} \in U_{\mathcal{M}}(\gamma), \quad i = 1, \cdots, n,$$

Let $\mathcal{P}_{\mathcal{M}^n}(\mathbf{x})^\top = [\mathcal{P}_{\mathcal{M}}(x_1)^\top, \cdots, \mathcal{P}_{\mathcal{M}}(x_n)^\top]$. By the definition of $\bar{\mathbf{x}}_{k+1}$ and Theorem 2, then we yield

$$\|\mathbf{x}_{k+1} - \bar{\mathbf{x}}_{k+1}\| \leq \|\mathbf{x}_{k+1} - \bar{\mathbf{x}}_k\|$$

$$= \left\| \mathcal{P}_{\mathcal{M}^n} \left( \mathbf{W}^t \mathbf{x}_k + \alpha_k \boldsymbol{\eta}_k \right) - \mathcal{P}_{\mathcal{M}^n}(\hat{\mathbf{x}}_k) \right\|$$

$$\leq \frac{1}{1-\gamma} \left\| \mathbf{W}^t \mathbf{x}_k + \alpha_k \boldsymbol{\eta}_k - \hat{\mathbf{x}}_k \right\| \tag{41}$$

$$\leq \frac{1}{1-\gamma} \sigma_2^t \|\mathbf{x}_k - \bar{\mathbf{x}}_k\| + \frac{2}{1-\gamma} \sqrt{n}\alpha_k L_g,$$

where the first inequality follows from the optimality of $\bar{\mathbf{x}}_{k+1}$, the second inequality uses Eq.(11), and the third inequality utilizes the fact that $\|\boldsymbol{\eta}_k\| \leq 2\sqrt{n}L_g$.

Let $\rho_t = \frac{1}{1-\gamma}\sigma_2^t$ where $0 < \rho_t < 1$, it follows from Eq.(42) that

$$
\begin{aligned}
\|\mathbf{x}_{k+1} - \bar{\mathbf{x}}_{k+1}\| &\leq \rho_t \|\mathbf{x}_k - \bar{\mathbf{x}}_k\| + \frac{2}{1-\gamma}\sqrt{n}\alpha_k L_g \\
&\leq \rho_t^{k+1} \|\mathbf{x}_0 - \bar{\mathbf{x}}_0\| + \frac{2\sqrt{n}L_g}{1-\gamma}\sum_{l=0}^{k}\rho_t^{k-l}\alpha_l.
\end{aligned}
\tag{42}
$$

Let $y_k = \frac{\|\mathbf{x}_k - \bar{\mathbf{x}}_k\|}{\sqrt{n}\alpha_k}$. For a positive integer $K \leq k$, it follows from Eq.(42) that

$$
y_{k+1} \leq \rho_t y_k + \frac{2}{1-\gamma}L_g \frac{\alpha_k}{\alpha_{k+1}} \leq \rho_t^{k+1-K}y_K + \frac{2}{1-\gamma}L_g \sum_{l=0}^{k}\rho_t^{k-l}\frac{\alpha_l}{\alpha_{l+1}}.
$$

Since $\alpha_k = \mathcal{O}(1/L_g)$ and $\|\bar{\mathbf{x}}_0 - \mathbf{x}_0\| \leq \frac{1}{2}\sqrt{n}\gamma$, one has that $y_0 \leq \frac{1}{2}\gamma/\alpha_0 = \mathcal{O}(L_g)$. Since $\lim_{k\to\infty}\frac{\alpha_{k+1}}{\alpha_k} = 1$, there exists sufficiently large $K$ such that $\alpha_k/\alpha_{k+1} \leq 2, \forall k \geq K$. For $0 \leq k \leq K$, there exists some $C' > 0$ such that $y_k^2 \leq C'\frac{1}{(1-\gamma)^2}L_g^2$, where $C'$ is independent of $L_g$ and $n$. For $k \geq K$, one has that $y_k^2 \leq C\frac{1}{(1-\gamma)^2}L_g^2$, where $C = 2C' + \frac{32}{(1-\rho_t)^2}$. Hence, we get $\frac{\|\mathbf{x}_k - \bar{\mathbf{x}}_k\|^2}{n} \leq C\frac{1}{(1-\gamma)^2}L_g^2$ for all $k \geq 0$, where $C = \mathcal{O}(\frac{1}{(1-\rho_t)^2})$. The proof is completed. $\qquad\square$

### D.1 PROOFS FOR THEOREM 1

*Proof.* We have the following inequality:

$$
\begin{aligned}
\|\operatorname{grad} & f(\bar{x}_k)\|^2 \\
&= \left\|\frac{1}{n}\sum_{i=1}^{n}\operatorname{grad} f_i(x_{i,k}) + \operatorname{grad} f(\bar{x}_k) - \frac{1}{n}\sum_{i=1}^{n}\operatorname{grad} f_i(x_{i,k})\right\|^2 \\
&\leq 2\left\|\frac{1}{n}\sum_{i=1}^{n}\operatorname{grad} f_i(x_{i,k})\right\|^2 + 2\left\|\operatorname{grad} f(\bar{x}_k) - \frac{1}{n}\sum_{i=1}^{n}\operatorname{grad} f_i(x_{i,k})\right\|^2 \\
&\leq 2\left\|\frac{1}{n}\sum_{i=1}^{n}\operatorname{grad} f_i(x_{i,k})\right\|^2 + \frac{2}{n}\sum_{i=1}^{n}\|\operatorname{grad} f_i(\bar{x}_k) - \operatorname{grad} f_i(x_{i,k})\|^2 \\
&\leq 2\left\|\frac{1}{n}\sum_{i=1}^{n}\operatorname{grad} f_i(x_{i,k})\right\|^2 + \frac{2L_g^2}{n}\sum_{i=1}^{n}\|\bar{x}_k - x_{i,k}\|^2 \\
&= 2\left\|\frac{1}{n}\sum_{i=1}^{n}\operatorname{grad} f_i(x_{i,k})\right\|^2 + \frac{2L_g^2}{n}\|\bar{\mathbf{x}}_k - \mathbf{x}_k\|^2,
\end{aligned}
\tag{43}
$$

where the third inequality uses Lemma 1. Since $\lim_{k\to\infty}\inf\|\operatorname{grad} f_i(x_{i,k})\| = 0$ based on Theorem 2 and $\|\bar{\mathbf{x}}_k - \mathbf{x}_k\|^2 \leq nC\frac{1}{(1-\gamma)^2}L_g^2\alpha_k^2$ based on Lemma 5, it follows from $\lim_{k\to\infty}\alpha_k = 0$ that

$$
\lim_{k\to\infty}\|\operatorname{grad} f(\bar{x}_k)\|^2 \leq 2\left\|\frac{1}{n}\sum_{i=1}^{n}\operatorname{grad} f_i(x_{i,k})\right\|^2 + 2C\frac{1}{(1-\gamma)^2}L_g^4\alpha_k^2 = 0.
$$

The proof is completed. $\qquad\square$

