# OpenReview forum: "Decentralized Riemannian Conjugate Gradient Method on the Stiefel Manifold"
_ICLR.cc/2024/Conference — ICLR 2024 poster_

### Official Review · Reviewer_yaab · 2023-10-23

**Soundness:** 3 good
**Presentation:** 2 fair
**Contribution:** 2 fair
**Rating:** 8
**Confidence:** 4

**Summary:**

The conjugate gradient method, a critical first-order optimization technique, typically exhibits faster convergence compared to the steepest descent method and demands significantly lower computational resources than second-order methods. Nevertheless, despite extensive research on various forms of conjugate gradient methods in Euclidean spaces and Riemannian manifolds, there has been limited exploration of such methods in distributed scenarios. This paper introduces a novel approach called the Decentralized Riemannian Conjugate Gradient Descent (DRCGD) method, designed to minimize a global objective function defined on the Stiefel manifold. This optimization problem is distributed among a network of agents, each associated with a local function, with communication occurring over a connected, undirected graph. Global convergence of DRCGD over the Stiefel manifold is proved. Numerical experiments demonstrate the advantages and efficacy of the DRCGD approach.

**Strengths:**

The paper is well written: basic definitions of optimization on Riemannian manifolds are recalled and the proposed algorithm is well explained. Furthermore, the latter is quite simple and hence has a great practical interest. Classical conjugate gradients on Riemannian manifolds have much faster convergence compared to the plain Riemannian gradient descent. Hence, proposing a decentralised Riemannian conjugate gradient descent can be well received by the community. Moreover, the global convergence of the proposed algorithm is proved whereas it is far from being trivial. Finally, numerical experiments show a practical interest to the proposed method.

The paper is overall of great quality.

**Weaknesses:**

The claim 2 in the introduction as well as the section 4.1 are misleading. Indeed, the authors mention they don't use retraction or vector transport to reduce the computation. However, they use the orthogonal projection onto the Stiefel manifold, which is a retraction, and the othogonal projection onto the tangent space, which is a vector transport (see "Optimization on matrix manifolds" from Absil et al. 2008). This claim should be removed.

In section 4.2, the equation (20) is not clear since $T_{alpha_k\eta_{i,k}^R$ is not defined. Hence, it is hard to appreciate if this hypothesis is reasonable or not. Same thing for assumption 3 (iii).

The paper lacks an overview of the poof to get the global convergence. The different proofs are long and technical and an overview would help the reader.

The numerical experiments section lacks the presentation of DRDGD and DPRGD. It would be interesting to better understand the differences with the proposed method.

Several passages in the proofs are unclear. See the questions.

**Questions:**

- Assumption 1: usually a doubly stochastic matrix is defined with positive elements and row and columns that sum to 1. Can you comment how does it relate to your definition?
- Section 4.1: "The Riemanian gradient step with a unit step size, i.e., ..." is it a unit step size or a null/zero step size?
- Assumption 3 (iii): what is $T_{alpha_k\eta_{i,k}^R$ ?

Proofs:
- Lemma 1, eq (24): how do you get second and third inequalities. For me, there is something wrong here.
- Theorem 2 is independent from the conjugate gradient. Is it new or is it a known result from a different paper?
- Theorem 2 assumes that $\eta_{i,k}=0$. In the proposed algorithm, you jointly do a gradient descent and average the iterates of the different nodes, hence $\eta_{i,k}\neq 0$. Can you comment this?
- After eq (31), you mention that $x_{i,k_0+1} \to x_{i,k_0}$. I don't understand at all this limit. Is it a mistake?
- Second inequality in eq (36), can you explain how do you get it?

Typos:
- Definition 3: (ii) $x_x$ is $0_x$.

Notations:
- $P_{St}$ and $P_M$ are the same.
- Section 4.2: $g_{i,k+1}$ is not introduced.

---

> ### Author Response · Authors · 2023-11-14
>
> Thank you for your valuable suggestions, and we have corrected the typos and inappropriate notations in the modified version.
> 1. [This claim 2 should be removed] Indeed, we don't use retraction directly. However, we still need to bound the projection operator using retraction and vector transport based on Eq.19 and Eq.20. Thus, we must give the complete claim of retraction and vector transport.
> 2. [$T_{\alpha_k \eta_{i,k}}^R$ is not defined] Actually, we give its definition in Example 1, which is the vector transport constructed by retraction.
> 3. [The paper lacks an overview of the poof] Thanks for your suggestion. We have added an overview of the proofs in Figure 5 in the modified version.
> 4. [The experiments lacks the presentation of DRDGD and DPRGD] In fact, the presentation of DRDGD and DPRGD is shown in Figure 4. And their results are very close.
> 5. [Doubly stochastic matrix] For example, if we have 4 agents with ER $p=0.3$ graph, then the matrix satisfies
> $W=\begin{bmatrix}
> 1/3  & 1/3 & 0 & 1/3  \\\
> 1/3  & 2/3 & 0 & 0 \\\
> 0 & 0 & 2/3 & 1/3 \\\
> 1/3 & 0 & 1/3 & 1/3
> \end{bmatrix} $. If we have 4 agents with Ring graph, then the matrix satisfies $W=\begin{bmatrix}
> 1/3  & 1/3 & 0 & 1/3  \\\
> 1/3  & 1/3 & 1/3 & 0 \\\
> 0 & 1/3 & 1/3 & 1/3 \\\
> 1/3 & 0 & 1/3 & 1/3
> \end{bmatrix}$.
> 6. [Is it a unit step size] It follows from the setting of DRDGD and DPRGD that a unit step size means the coefficient before $\sum_{j=1}^n W_{i j}^t x_{j,k}$ is 1.
> 7. [Assumption 3 (iii): what is $T_{\alpha_k \eta_{i,k}}^R$] It is a vector transport constructed by retraction, which is defined in Example 1.
> 8. [Lemma 1, eq (24): how do you get second and third inequalities] We miss a parenthesis in the second inequality, which has been corrected in the modified version. And $0 \leq \Vert x \Vert, \Vert y \Vert \leq 1$ is implicit in this proof.
> 9. [Is Theorem 2 new or is it a known result from a different paper] In fact, similar results have been given in [1], which we extended from retraction to projection operator and made it compatible with the conjugate gradient.
> 10. [Theorem 2 assumes that $\eta_{i,k}=0$] It's fine. Theorem 2 discusses the consensus error $\sum_{j=1}^n W_{i j}^t x_{j,k}$, and $\eta_{i,k}$ does not belong to the consensus error, so we consider setting it to 0. In particular, we still consider the role of $\eta_{i,k}$ in the part of global convergence (Appendix D).
> 11. [$x_{i,k_0+1} \rightarrow x_{i,k_0}$] There is no problem. It means that $x_{i,k_0+1}$ will no longer be updated and will remain at the previous point $x_{i,k_0}$ as the consensus error converges and the gradient approaches 0. A similar statement can be found in Theorem 4.2 based on [2].
> 12. [Second inequality in eq (36)] First, the second part comes from Eq. 35. Then, the third part is $$ \Vert \mathcal{P}\_{T\_{x\_{i,k}}\mathcal{M}}\left(\sum\_{j=1}\^n W\_{i j}\^t \eta\_{j,k-1}\right) \Vert \_{x\_{i,k}}\^2  \leq \Vert \sum\_{j=1}\^n W\_{i j}\^t \eta\_{j,k-1} \Vert \_{x\_{i,k}}\^2  \leq \Vert  \eta\_{i,k-1} \Vert \_{x\_{i,k-1}}\^2$$, where $ \Vert \sum\_{j=1}\^n W\_{i j}\^t \eta\_{j,k-1} \Vert \_{x\_{i,k}}\^2  \leq \Vert  \eta\_{i,k-1} \Vert \_{x\_{i,k-1}}\^2$ is a similar assumption followed with formula (4.28) in [2].
>
> [1] Chen, Shixiang, et al. "Decentralized riemannian gradient descent on the stiefel manifold." International Conference on Machine Learning. PMLR, 2021.
>
> [2] Sato, Hiroyuki. Riemannian optimization and its applications. Berlin: Springer, 2021.

---

> > ### Comment · Reviewer_yaab · 2023-11-14
> >
> > I thank the authors for the answers to my questions.
> > I still have two concerns:
> >
> > question 8: why can you assume that $0 \leq ||x||, ||y|| \leq 1$ ?
> >
> > question 12: in [2], eq 4.28 is explicitly stated in the assumptions. Maybe you should do so in your theorem?

---

> > > ### Author Response · Authors · 2023-11-14
> > >
> > > 1. $0 \leq \Vert x \Vert, \Vert y \Vert \leq 1$ is not an assumption. $x$ and $y$ are obviously bounded due to $x,y \in \operatorname{St}(d,r)$. However, whatever this upper bound is, we can normalize it, where the normalization coefficient can be incorporated into $L_f$. So, this is just a convenient description after normalization.
> > > 2. This is indeed an omission, and we have added a corresponding statement in Theorem 3. Thank you again for your valuable suggestions on our paper.

---

> > > > ### Comment · Reviewer_yaab · 2023-11-14
> > > >
> > > > Thank you for the clarification.
> > > >
> > > > The authors answered my concerns and I raised the rating from 6 to 8.

---

### Official Review · Reviewer_86eb · 2023-10-31

**Soundness:** 4 excellent
**Presentation:** 3 good
**Contribution:** 3 good
**Rating:** 6
**Confidence:** 4

**Summary:**

This paper suggests an extension of conjugate gradient on Stiefel manifold for distributed setting and provides convergence guarantees for this algorithms. The approach is based mainly on the Xiaojing Zhu's original papers

**Strengths:**

The setting of decentralized optimisation is crucial in applied optimisation, and extending one of the most practically efficient algorthms to it is topical. Riemannian generality here requires carefull trheoretical justifications and proper choice of tools to prevent big computational complexity. Paper indeed propose a good solution for solving optimisation problems on Stiefel manifold.

**Weaknesses:**

Empirical study is not comprehensive: there was not presented a comparison of the proposed approach with alternatives. Besides, form of convergence guarantees is not exhaustive, because the rate of the convergence is not established. Theoretical framework is mostly inherited from Zhu's original papers, but that analysis does not allow providing guarantees on convergence rate, so does not this paper, which means that there were no significant extending of that framework.

**Questions:**

1. Typo in "Lemma 3 In Alogrithm"
2. What about time-varying case? Conidering the case of time-varying graph would be important for all-around extending CG for decentralised setting.

---

> ### Author Response · Authors · 2023-11-14
>
> Thank you for your efforts in improving the quality of our paper. In fact, in the framework of conjugate gradient, the convergence rate has not been established so far. Naturally, the decentralized version we derive from this framework can not provide the convergence rate as well. We guess the view inherited from Zhu's original papers is caused by the framework of the Riemannian conjugate gradient. Different studies revolve around geometric tools without affecting the theoretical framework of conjugate gradient, e.g., inverse retraction [1] and vector transport [2]. Specifically, we present a quite simple geometric tool (projection operator), and extend it to decentralized scenarios, which obviously makes a significant extension to unify the framework of conjugate gradient and decentralized optimization.
> 1. We have modified the corresponding typo in the modified version.
> 2. This is a very meaningful and interesting idea for the time-varying graph in decentralized CG. We also notice that there have been some recent works focusing on time-varying graphs, e.g., [3], [4], [5]. By reviewing these works, we find that this paper can be extended to time-varying cases because we have no additional requirements for decentralized settings. Of course, there are a large number of proofs to be considered that would go well beyond the scope of this paper and would be sufficient to form a new paper. We will consider this exciting setting in future work.
>
> [1] Zhu, Xiaojing, and Hiroyuki Sato. "Riemannian conjugate gradient methods with inverse retraction." Computational Optimization and Applications. 2020.
>
> [2] Sato, Hiroyuki, and Toshihiro Iwai. "A new, globally convergent Riemannian conjugate gradient method." Optimization. 2015.
>
> [3] Rogozin, Alexander, et al. "An accelerated method for decentralized distributed stochastic optimization over time-varying graphs." IEEE Conference on Decision and Control (CDC). 2021.
>
> [4] Saadatniaki, Fakhteh, Ran Xin, and Usman A. Khan. "Decentralized optimization over time-varying directed graphs with row and column-stochastic matrices." IEEE Transactions on Automatic Control. 2020.
>
> [5] Kovalev, Dmitry, et al. "ADOM: Accelerated decentralized optimization method for time-varying networks." International Conference on Machine Learning. 2021.

---

### Official Review · Reviewer_hbTT · 2023-11-02

**Soundness:** 2 fair
**Presentation:** 2 fair
**Contribution:** 3 good
**Rating:** 5
**Confidence:** 3

**Summary:**

This paper presents a decentralized Riemannian conjugate gradient descent (DRCGD) algorithm for distributed optimization, and proves the global convergence of the algorithm. Compared with existing state-of-the-art algorithms, DRCGD uses a projection operator that searches the direction instead of retraction and vector transport, thus reducing computational costs. Through the simulation of eigenvalue problem, the paper shows that DRCGD has better performance than state-of-the-art algorithms.

**Strengths:**

originality and significance: This paper has good originality and significance since it presents the first decentralized Riemannian conjugate gradient descent (DRCGD) algorithm for distributed optimization and proves the global convergence of the algorithm.

quality: The proposed algorithm is supported by solid theory and verified by simulation.

clarity: The overall narrative logic of the article is clear.

**Weaknesses:**

1. There is still room for improvement in the clarity of the proof. Some symbols that appear in the convergence analysis section of the text, such as $g_{i,k+1}$, $\mathcal{N}$, and $C$, are not defined in the text.

2. It seems that there are some assumptions about the step size $\alpha_{k}$ that are not mentioned in Assumption 3 about $\alpha_{k}$ in the body of the proof, such as the assumption about $\alpha_{k}$ in Lemma 2. This leads to unclear assumptions about $\alpha_{k}$.

3. The measures mentioned in the simulation should converge towards 0, which is not well demonstrated in the experimental results. For example, the measures in Figure 3 tend to be constant after it drops to a certain level. This does not support the theoretical results very well.

4. The definition of doubly stochastic matrix seems to be $\sum_{i} x_{ij}=\sum_{i} x_{ij}=1$, which is different from the definition in Assumption 1.

**Questions:**

1. What is the definition of $x_{x}$ in Definition 3 (ii)?

2. It seems like there are many assumptions of the parameters such as $\alpha_{k}$. Are these assumptions easy to satisfy?

3. In this paper, the decreasing step size is used in the convergence proof, while the fixed step size is used in the simulation. Does this difference affect convergence?

4. What is the significance of the eigenvalue problem used in the simulation in real life?

5. In general, since the problems solved are the same, the convergence result of the distributed algorithm should be independent of the structure of the graph if the assumptions about the graph are satisfied. In Figure 3 of the simulation, the same algorithm seems to converge to different solutions under different graphs. Why did this happen?

---

> ### Author Response · Authors · 2023-11-14
>
> 1. [Some symbols are not defined] Thank you for your valuable suggestions, and we have added the definitions of the corresponding symbols in the modified version.
> 2. [Some assumptions about step size are not mentioned] Indeed, the step size $\alpha_k$ should be bounded as the description of Lemma 2, and we have added this part to Assumption 3.
> 3. [The measures tend to be constant] In theoretical analysis, we require the step size to be infinitely close to 0. However, for a fair comparison with different methods in the experiment, we employ fixed step sizes, i.e., $\alpha_k=\frac{\hat{\alpha}}{\sqrt{K}}$ with $K$ being the maximal number of iterations. Obviously, this causes the measures only drop to a certain level.
> 4. [Doubly stochastic matrix] In our opinion, the definition in Assumption 1 puts forward higher requirements for a doubly stochastic matrix, including symmetry and eigenvalues, but it still meets the properties of a doubly stochastic matrix. Actually, the completely consistent definition is also given in Assumption 1 based on [1].
> 5. [Definition of $x_x$ in Definition 3 (ii)] It should be $0_x$, and we have corrected it in the modified version.
> 6. [Are these assumptions easy to satisfy] Many assumptions are built into conjugate gradient, e.g., the Armijo condition and strong Wolfe condition, and their purpose is to theoretically prove the convergence of conjugate gradient. All assumptions only ensure that the algorithm converges in the worst case, and they can be satisfied from the view of practical convergence.
> 7. [Does this difference affect convergence] The fixed step size causes the measures approach to be constant (very close to 0) after it drops to a certain level, rather than continuously dropping as the decreasing step size.
> 8. [The eigenvalue problem in real life] The eigenvalue problem is a fundamental concept in linear algebra with widespread applications in various fields, including physics, engineering, computer science, and statistics. In the context of simulations and real-life applications, the eigenvalue problem is particularly significant for several reasons: (1) Eigenvalues are used in control theory to analyze the stability of control systems. (2) Eigenvalue decomposition is utilized in various algorithms for dimensionality reduction and feature extraction in machine learning. (3) Eigenvalue problems are foundational in spectral methods used in numerical simulations.
> 9. [The same algorithm seems to converge to different solutions] In practice, the proposed method indeed achieves convergence under different graphs, which shows that their convergence is indeed independent of the structure of the graph. However, different graph structures will cause different efficiency of information transmission, thus affecting the convergence trend.
>
> [1] Chen, Shixiang, et al. "Decentralized riemannian gradient descent on the stiefel manifold." International Conference on Machine Learning. PMLR, 2021.

---

### Comment · Area_Chair_Lje3 · 2023-11-20
**Discussion phase closing soon**

Dear authors, dear reviewers,

As a reminder, the author-reviewer discussion period will be coming to close in about two days.

To make sure that this phase is as constructive as possible, I would kindly ask the reviewers, if you haven’t already done so, to go through the authors’ posted rebuttals, follow up on their replies to your comments, and engage with the authors if you would like to ask any further questions.

Once the discussion period closes, it will be harder to get input from the authors, so it would be better to do this before the last day.

Regards,

The AC

---

### Meta-Review · Area_Chair_Lje3 · 2023-12-06

**Metareview:**

This paper presents a version of the conjugate gradient method adapted to Riemannian optimization problems over Stiefel manifolds. The setting considered is that of distributed optimization, whereby a set of agents/workers connected via a communication graph seek to optimize a finite-sum objective in a decentralized manner. The authors' main contribution is a retraction-based algorithm based on the conjugate gradient idea, which is shown to possess a convergent subsequence.

While the algorithm's theoretical convergence guarantees are not the strongest one could hope for (they concern a specific manifold and the $\liminf$ of the Riemannian gradient norm, so they imply asymptotic convergence of a subsequence but no more than that), the problem itself is technically quite challenging and difficult, and the provided simulations encouraging.

Overall, the discussion with the reviewers veered toward the positive, with two concerns remaining:
- In Figs. 1 and 3, the gradient norm converges to a constant. The authors explained this as follows: in theory, the algorithm uses a series of step sizes that gradually approach zero, while in simulation, a fixed step size is selected for comparison with the benchmark. It would be useful if the authors provided simulations using a series of step sizes that gradually approach zero, or by showing convergence results for different (fixed) step sizes.
- Different graphs may lead to different information-passing efficiencies, which can result in different numbers of epochs needed to achieve the same error. This seems to contradict Fig. 3, a fact which should also be discussed.

Conditioned on addressing the above issues, the committee recommends "acceptance".

**Justification For Why Not Higher Score:**

The scope of the paper (distributed optimization over the Stiefel manifold) is somewhat narrow.

**Justification For Why Not Lower Score:**

The contribution of the paper is solid.

---

### Decision · Program_Chairs · 2024-01-16

Accept (poster)